# Antiferromagnetic textures in BiFeO$_3$ controlled by strain and electric field

A. Haykal[1,9], J. Fischer [2,9], W. Akhtar[1,8], J.-Y. Chauleau[3], D. Sando [4], A. Finco [1], F. Godel [2], Y. A. Birkhölzer [5], C. Carrétéro[2], N. Jaouen [6], M. Bibes [2], M. Viret [3], S. Fusil[2,7 ✉], V. Jacques[1] & V. Garcia [2]

Antiferromagnetic thin films are currently generating considerable excitement for low dissipation magnonics and spintronics. However, while tuneable antiferromagnetic textures form the backbone of functional devices, they are virtually unknown at the submicron scale. Here we image a wide variety of antiferromagnetic spin textures in multiferroic BiFeO$_3$ thin films that can be tuned by strain and manipulated by electric fields through room-temperature magnetoelectric coupling. Using piezoresponse force microscopy and scanning NV magnetometry in self-organized ferroelectric patterns of BiFeO$_3$, we reveal how strain stabilizes different types of non-collinear antiferromagnetic states (bulk-like and exotic spin cycloids) as well as collinear antiferromagnetic textures. Beyond these local-scale observations, resonant elastic X-ray scattering confirms the existence of both types of spin cycloids. Finally, we show that electric-field control of the ferroelectric landscape induces transitions either between collinear and non-collinear states or between different cycloids, offering perspectives for the design of reconfigurable antiferromagnetic spin textures on demand.

[1] Laboratoire Charles Coulomb, Université de Montpellier and CNRS, 34095 Montpellier, France. [2] Unité Mixte de Physique, CNRS, Thales, Université Paris-Saclay, 91767 Palaiseau, France. [3] SPEC, CEA, CNRS, Université Paris-Saclay, 91191 Gif-sur-Yvette, France. [4] School of Materials Science and Engineering, University of New South Wales, Sydney 2052, Australia. [5] Department of Inorganic Materials Science, Faculty of Science and Technology and MESA+ Institute for Nanotechnology, University of Twente, P.O. Box 217, 7500 AE Enschede, The Netherlands. [6] Synchrotron SOLEIL, 91192 Gif-sur-Yvette, France. [7] Université d'Evry, Université Paris-Saclay, Evry, France. [8] Present address: Department of Physics, JMI, Central University, New Delhi, India. [9] These authors contributed equally: A. Haykal, J. Fischer. ✉email: stephane.fusil@cnrs-thales.fr

In ferromagnetic materials, spin textures are conventionally tweaked with a magnetic field. Antiferromagnetic spin textures, on the other hand, are intrinsically insensitive to external magnetic fields, calling for alternative control knobs to manipulate the antiferromagnetic order. The electrical manipulation of antiferromagnetism was recently demonstrated in non-centrosymmetric metallic antiferromagnets[1–3]; however, the spin orbit torque required to either switch by 90° or reverse by 180° the antiferromagnetic vector involves large current densities of the order of $10^6$–$10^7$ A cm$^{-2}$. Furthermore, the efficiency of this writing method faces limitations, since only a small fraction of antiferromagnetic domains is actually switched[4,5]. An optimal writing mechanism would demand low current densities (or ideally no current) to generate a complete reversal of anti-ferromagnetic domains or textures. Recent reports have for instance demonstrated that piezoelectric strain can provide low power control of antiferromagnetic memories[6,7].

In some materials possessing both antiferromagnetic and electrical orders, the magnetoelectric coupling is an additional means expected to efficiently channel electric-field stimuli onto the antiferromagnetic order. Yet, the fundamental ingredients deterministically governing the imprint of the ferroelectric order to the antiferromagnetic order remain poorly understood. Even in the archetypal room-temperature multiferroic[8], BiFeO$_3$, the details of the antiferromagnetic textures are virtually unknown at the scale of ferroelectric domains. The seminal work of Zhao et al. showed promise for the electric control of the antiferromagnetic order in BiFeO$_3$ thin films[9]. To date, its complex anti-ferromagnetic order has been solely inferred from volume averaged techniques such as neutron diffraction, Mössbauer spectroscopy, or Raman spectroscopy. Depending on the strain, growth conditions and crystal orientation, the magnetic state of BiFeO$_3$ thin films can either show different types of non-collinear cycloids, canted G-type antiferromagnetic orders, or even a mixture of these[10,11]. More generally, examples of anti-ferromagnetic textures being imaged at the nanoscale are extremely scarce in the literature[12–14]. Here we bring deep insight into the strain-dependent interplay between the ferroelectric and antiferromagnetic orders at the local scale and show that electric field can be used to convert between various collinear and non-collinear spin arrangements.

## Results

### Strain-engineered BiFeO$_3$ with striped ferroelectric domains.
BiFeO$_3$ thin films were grown using pulsed laser deposition on various substrates (SrTiO$_3$, DyScO$_3$, TbScO$_3$, GdScO$_3$, SmScO$_3$) with a thin bottom electrode of SrRuO$_3$ (Methods). X-ray dif-fraction shows the high epitaxial quality of the films with Laue fringes (Fig. 1a–e) attesting for their coherent growth. All films display smooth surfaces with atomic steps, characteristic of a layer-by-layer growth (insets of Fig. 1a–e). The (001) BiFeO$_3$ peak evolves from the left to the right of the substrate (001) peak upon increase of the in-plane pseudo-cubic lattice parameter of the substrate, as observed in the $2\theta$–$\omega$ scans. Reciprocal space maps indicate that the films are fully strained (Supplementary Fig. 1) with only two elastic variants of the BiFeO$_3$ monoclinic phase (Fig. 1f–j). Their peak positions enable us to determine a strain value for each film ranging from −1.35% compressive strain to +0.50% tensile strain (Fig. 1k, Supplementary Fig. 1 and Methods).

With this set of structurally equivalent BiFeO$_3$ thin films, distinguishable only by their strain level, we now focus on the evolution of the ferroelectric and magnetic textures (Fig. 2). In BiFeO$_3$, the displacement of Bi ions relative to the FeO$_6$ octahedra gives rise to a strong ferroelectric polarisation along one of the

<111> directions of the pseudo-cubic unit cell. The out-of-plane and in-plane variants of polarisation were identified in each sample using piezoresponse force microscopy (PFM; Methods). For all the samples, the as-grown out-of-plane polarisation is pointing downward, i.e. towards the bottom electrode (Supplementary Fig. 2a). Figure 2a–e displays similar striped-domain structures with two in-plane ferroelectric variants, which correspond to the two elastic domains observed in reciprocal space maps[15]. In contrast to the as-grown striped domain patterns of the BiFeO$_3$ films grown on the scandates, the striped domain pattern of the BiFeO$_3$ film on SrTiO$_3$ was defined by PFM (Supplementary Fig. 4). All the samples can be considered as a periodic array of 71-degree domain walls, separated by two ferroelectric variants (Supplementary Figs. 2 and 3). This ordered ferroelectric landscape greatly simplifies the exploration and interpretation of the magnetic configuration for each ferroelectric domain[16].

### Influence of the strain on the antiferromagnetic textures.
For each sample, the corresponding antiferromagnetic spin textures were imaged in real space with a scanning NV (nitrogen-vacancy) magnetometer[17] operated in dual-iso-**B** imaging mode (Fig. 2g–k, Methods, Supplementary Fig. 5). In the strain range of −1.35 to +0.05%, the NV images display a similar zig-zag pattern of periodic stray fields generated by cycloidal antiferromagnetic orders. More precisely, in each vertical ferroelectric domain (separated by dashed lines in Fig. 2g–j), we observe a single propagation direction of the spin cycloid. As the in-plane variant of polarisation rotates from one domain to another, the spin cycloid propagation direction rotates accordingly. This implies a one-to-one correspondence between the ferroelectric and anti-ferromagnetic domains. In contrast, for large tensile strain (+0.5%) corresponding to BiFeO$_3$ films grown on SmScO$_3$ sub-strates, the cycloidal order appears to be strongly destabilized (Fig. 2k and Supplementary Fig. 6). In this specific case, the ferroelectric periodicity is lost in the magnetic pattern, which may suggest a weaker magnetoelectric coupling as compared to other magnetic interactions. This strain dependence of the magnetic textures is reminiscent of previous works where anti-ferromagnetic order as a function of strain was studied by non-local techniques such as Mössbauer and Raman spectro-scopies[10,11]. Indeed, a canted G-type antiferromagnetic order was identified for tensile strain over +0.5% and a cycloidal order from −1.6% to +0.5%.

In the present sample set, the magnetic image of BiFeO$_3$ films grown on DyScO$_3$ substrates (Fig. 2h) with −0.35% strain corresponds to the configuration already observed by Gross et al.[16]. The 90-degree in-plane rotation of the ferroelectric polarisation imprints the 90-degree in-plane rotation of the cycloidal propagation direction. This corresponds to one of the three bulk-like cycloids (cycloid I) with propagation vectors contained in the (111) plane orthogonal to the polarisation[18,19] (Fig. 3a, b). Among them, the observed $\mathbf{k}_1$ vector lies in the (001) plane of the film, for both ferroelectric variants (Fig. 2h). For lower compressive strain (−0.10%, TbScO$_3$), the magnetic configuration is found to be identical (Fig. 2i), also corresponding to the bulk-like cycloid (cycloid I, $\mathbf{k}_1$).

A subtle change of the strain towards the tensile side (+0.05%, GdScO$_3$) greatly influences the magnetic landscape. Indeed, the spin texture can no longer be explained by the bulk-like cycloid as the zig-zag features are no longer orthogonal to each other, but rather at $120 \pm 5$ degrees (Fig. 2j). Interestingly, for (001) BiFeO$_3$ films grown under low tensile strain (+0.2%), previous reports have shown evidence for exotic spin cycloids[10,11]. In these works, Mössbauer and nuclear resonant scattering data suggested

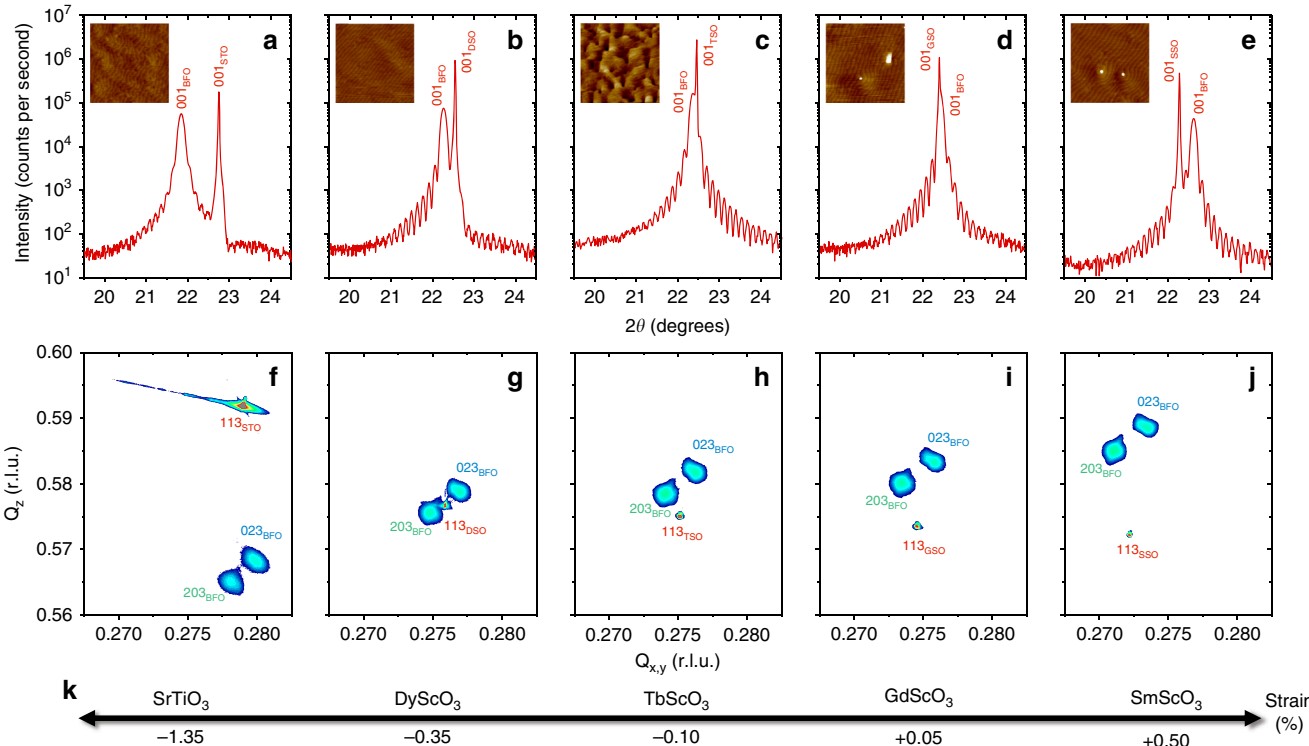

**Fig. 1 Strain-engineered epitaxial BiFeO₃ thin films. a–e**, $2\theta$-$\omega$ X-ray diffraction scans of BiFeO₃ (BFO) films grown on SrTiO₃ (STO) (**a**), DyScO₃ (DSO) (**b**), TbScO₃ (TSO) (**c**), GdScO₃ (GSO) (**d**) and SmScO₃ (SSO) (**e**) substrates. The insets are $3 \times 3\,\mu m^2$ topography images acquired by atomic-force microscopy on the same films, showing atomic steps and terraces. The z-scale is 4 nm. **f–j** Corresponding reciprocal space maps along the different (113) substrate peaks, showing in each case two elastic domains for BiFeO₃, i.e. (203) and (023). The r. l. u. units of the in-plane and out-of-plane wavevectors, $Q_{x,y}$ and $Q_z$, respectively, stand for reciprocal lattice units. **k** Sketch of the evolution of the calculated epitaxial strain in BiFeO₃ as a function of the substrate. The scandate and BiFeO₃ crystallographic peaks are defined in a monoclinic cell.

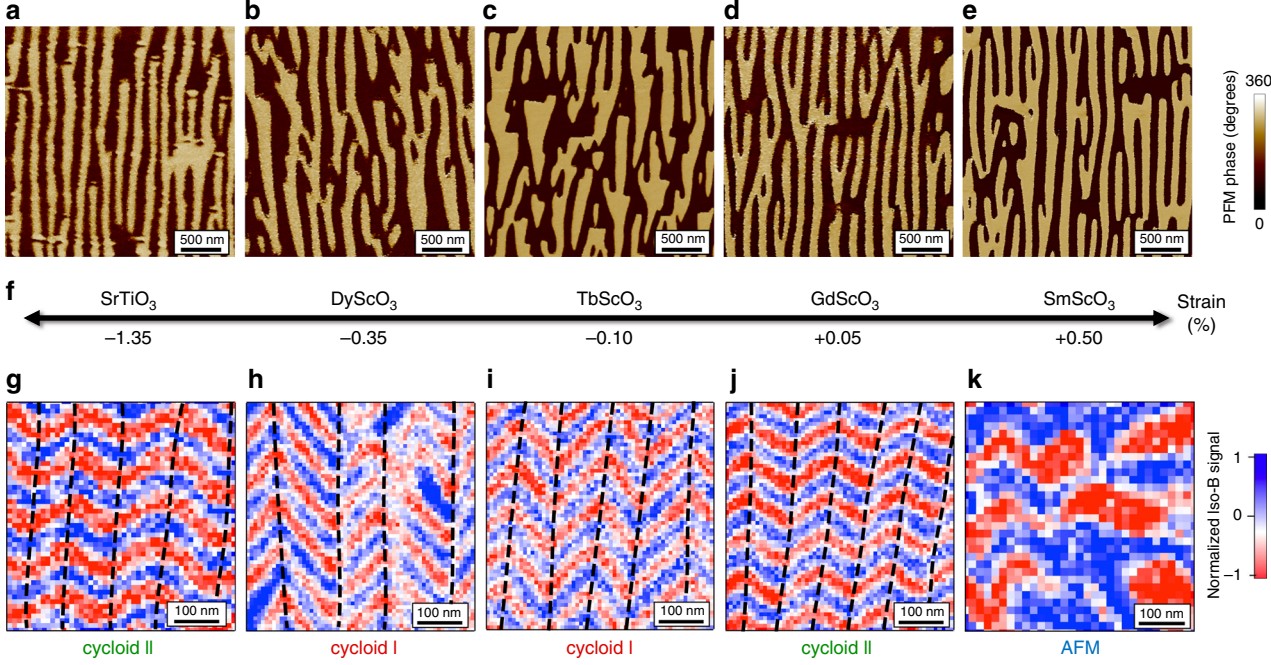

**Fig. 2 Strain dependent magnetic textures on striped ferroelectric domains. a–e** In-plane PFM phase images of BiFeO₃ films grown on SrTiO₃ (**a**), DyScO₃ (**b**), TbScO₃ (**c**), GdScO₃ (**d**) and SmScO₃ (**e**) substrates. **f** Sketch of the evolution of the epitaxial strain in BiFeO₃ as a function of the substrate. **g–k** NV magnetometry images corresponding to the ferroelectric domains depicted in (**a–e**). The dashed lines in (**g–j**) are guides to the eyes, reflecting a change of the cycloid propagation vector associated to the ferroelectric domain walls. The symbol AFM in (**k**) stands for pseudo-collinear antiferromagnetic order.

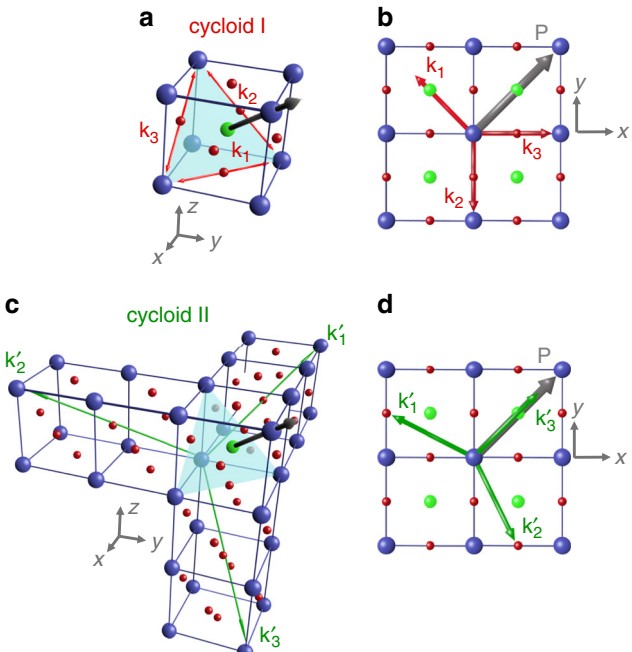

**Fig. 3 Sketches of the different types of spin cycloids in BiFeO₃. a, b** Bulk-like spin cycloid (cycloid I) with the three possible propagation vectors for each polarisation variant in 3D view (**a**) and top view (**b**). **c, d** The exotic spin cycloid (cycloid II) with propagation vectors along the three <11$\bar{2}$> directions in 3D view (**c**) and top view (**d**).

a propagation vector contained in the ($\bar{1}$10) plane[10,11]. This result was recently supported by neutron diffraction experiments on Co-doped BiFeO₃ films grown on SrTiO₃(110), where the propagation vector of the spin cycloid was found to be along the [11$\bar{2}$] direction[20]. Guided by these observations, here we consider three possible propagation directions ($k_1'$, $k_2'$, $k_3'$) for the cycloid II; namely along [$\bar{2}$11], [1$\bar{2}$1], and [11$\bar{2}$], respectively (Fig. 3c, d). In the case of BiFeO₃ thin films on GdScO₃ substrates (Fig. 2j), the angle of the zig-zag pattern is only compatible with alternating $k_1'$, $k_2'$ propagation vectors, giving rise to an angle of 127 degrees, as projected on the film surface. Surprisingly, a similar scenario takes place for large compressive strain (−1.35%, SrTiO₃) as the zig-zag angle (Fig. 2g) is the same as for BiFeO₃ grown on GdScO₃. This unprecedented real-space observation of the cycloid II under both large compressive strain and low tensile strain calls for further theoretical input to explain the interplay between strain and antiferromagnetic textures.

**Insights into the different spin cycloids**. To further corroborate the nanoscale real-space images of the magnetic arrangements, complementary macroscopic investigations were performed by X-ray resonant elastic scattering on BiFeO₃ samples[21,22] grown on both DyScO₃ (cycloid I) and GdScO₃ (cycloid II) substrates (Fig. 4a, c). As the spin cycloid is a periodic magnetic object, it gives rise to a diffracted pattern at the Fe resonant *L*-edge. In order to select the diffracted signal of magnetic origin, the difference between left and right circularly polarized light is plotted as a dichroic diffracted pattern (Fig. 4a, red and blue correspond to positive and negative dichroism, respectively). In both diagonals from the specular spot, the inverted contrast between +**q** and −**q** spots is a signature of chirality. Indeed, BiFeO₃ spin cycloids in which spins rotate in a plane defined by the polarisation (**P**) and the propagation vector (**k**) are chiral objects.

For BiFeO₃ thin films grown on DyScO₃, the presence of two orthogonal cycloid propagation directions (red arrows in Fig. 4a)

with identical periods gives rise to two orthogonal lines of diffracted spots, thus defining a square diffracted pattern. The fine structure of this pattern is rendered more complex by additional spots that arise from the modulation of the magnetic periodicity by the ferroelectric domain structure;[23] however, here our focus is on the cycloid propagation direction and periodicity. The spacing between the +**q** and −**q** spots corresponds to a cycloid period of 72 ± 5 nm for both spin cycloids with $k_1$ propagation vector. Consistently at the local scale, the combination of PFM and scanning NV magnetometry allows to identify the relative orientation of the ferroelectric polarisation (**P**, grey arrows in Fig. 4b) and cycloid propagation direction ($k_1$, red arrows in Fig. 4b) on both sides of a domain wall. Thus, our microscopic real-space experiments and macroscopic reciprocal-space observations both attest for a single cycloidal vector ($k_1$) in BiFeO₃ thin films under moderate compressive strain.

In contrast, for BiFeO₃ films grown on GdScO₃ imposing slight tensile strain, the dichroic diffracted pattern is no longer square but rectangular (Fig. 4c). Hence, we preclude the above-mentioned scenario with two bulk-like (cycloid I) orthogonal vectors. The two diagonals of the rectangular pattern (green arrows in Fig. 4c) form an angle of about 110 ± 5 degrees, in accordance with the typical angles observed in NV magnetometry images. The only plausible scenario, therefore, corresponds to two types of ferroelectric domains respectively harbouring alternating $k_1'$ and $k_2'$ propagation vectors of the cycloid II, as observed in real space (Fig. 4d). These two cycloid propagation variants appear to be energetically degenerated and favoured over the more out-of-plane $k_3'$ vector (Fig. 3c). Consequently, these cycloidal BiFeO₃ films, under either compressive or tensile strain, exhibit a one-to-one imprint between ferroelectric and antiferromagnetic order.

**Electric-field control of antiferromagnetic textures**. Beyond the observations on pristine configurations of ferroelectric domains in which the cycloid propagation is locked onto the polarisation, we now manipulate the ferroelectric order using electric fields, with the aim to design antiferromagnetic landscapes on demand. We first use PFM to draw micron-size ferroelectric domains (Supplementary Fig. 7) by virtue of the so-called trailing field[24–26]. Using microdiffraction experiments, we checked that no strain difference could be detected between artificially written and as-grown striped-domains (Methods and Supplementary Fig. 8). NV magnetometry is then performed on these artificial domains to reveal the corresponding magnetic textures (Fig. 5 and Supplementary Fig. 7). For strain states ranging from −0.35 to +0.50%, single ferroelectric domains always correspond to a spin cycloid with a single propagation vector. For BiFeO₃ films grown on DyScO₃ (−0.35%, Fig. 5a) or TbScO₃ (−0.10%, Fig. 5b), the spin cycloid propagates in a direction perpendicular to the ferroelectric polarisation. This implies that the in-plane $k_1$ propagation is still favoured, switching from two pristine cycloid Is to a single written cycloid I. Interestingly, the spin cycloid period λ decreases from about 78 ± 5 nm in the pristine (two domain) state to 65 ± 2 nm for the switched (single domain) state. In single domains, the spin cycloid period thus appears closer to that observed in bulk BiFeO₃ (λ_bulk = 64 nm, ref.[19]), suggesting that periodic electric/elastic boundary conditions influence the cycloid period.

For BiFeO₃ films grown on GdScO₃ (+0.05%, Fig. 5c), the spin cycloid propagates horizontally, i.e. at 45 degrees from the in-plane polarisation variant of the single ferroelectric domain. This implies that the cycloid I out-of-plane propagation vector ($k_2$, Fig. 3a, b) is selected, corresponding to a switching from two cycloid IIs ($k_1'$, $k_2'$) to a single cycloid I ($k_2$). In addition, the apparent cycloid period of 92 ± 3 nm in the single domain is compatible with its projection onto the sample surface

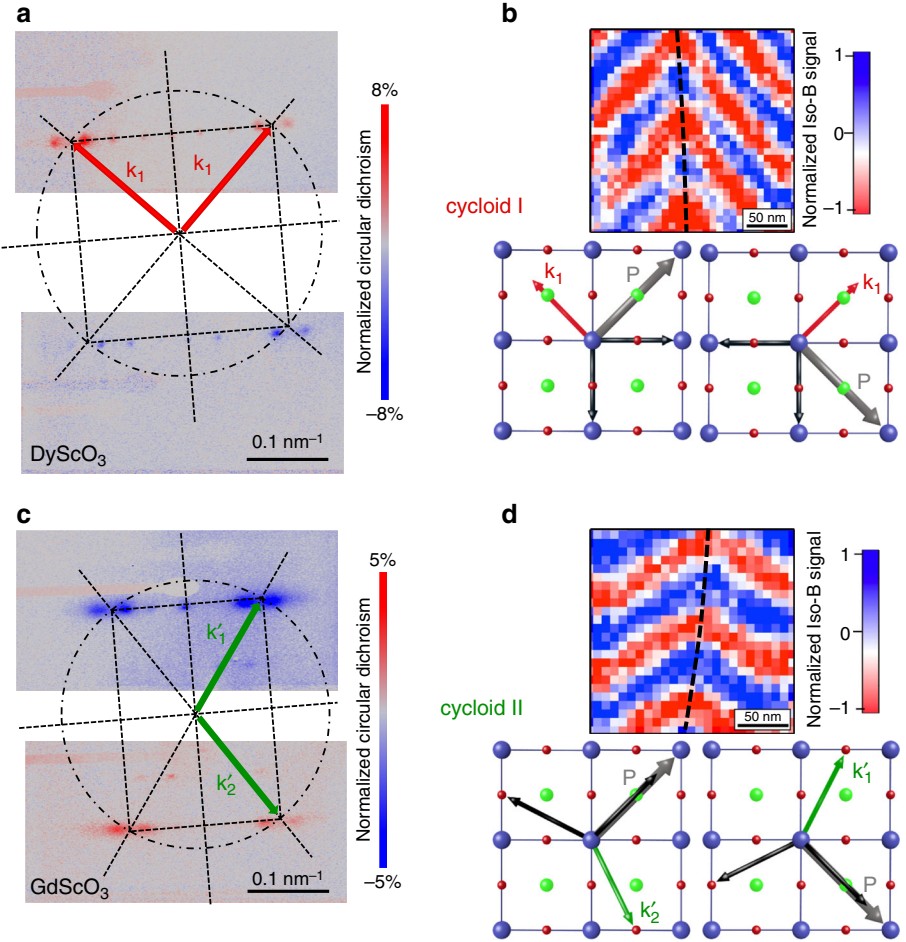

**Fig. 4 The two types of spin cycloids in real and reciprocal spaces. a** Resonant X-ray elastic scattering at the Fe *L*-edge for BiFeO$_3$ grown on DyScO$_3$. The square pattern indicates a bulk-like cycloid (cycloid I) with propagation vectors aligned 90 degrees from each other. **b** Corresponding NV magnetometry image zoomed in, with the propagation vectors sketched for both polarisation variants. **c** Resonant X-ray elastic scattering at the Fe *L*-edge for BiFeO$_3$ grown on GdScO$_3$. The rectangular pattern corresponds to the cycloid II with propagation vectors lying at 110 ± 5 degrees from each other. **d** Corresponding NV magnetometry image zoomed in, with the propagation vectors sketched for both polarisation variants. The dashed lines in **b**, **d** are guides to the eyes, reflecting a change of the cycloid propagation vector associated to the ferroelectric domain walls.

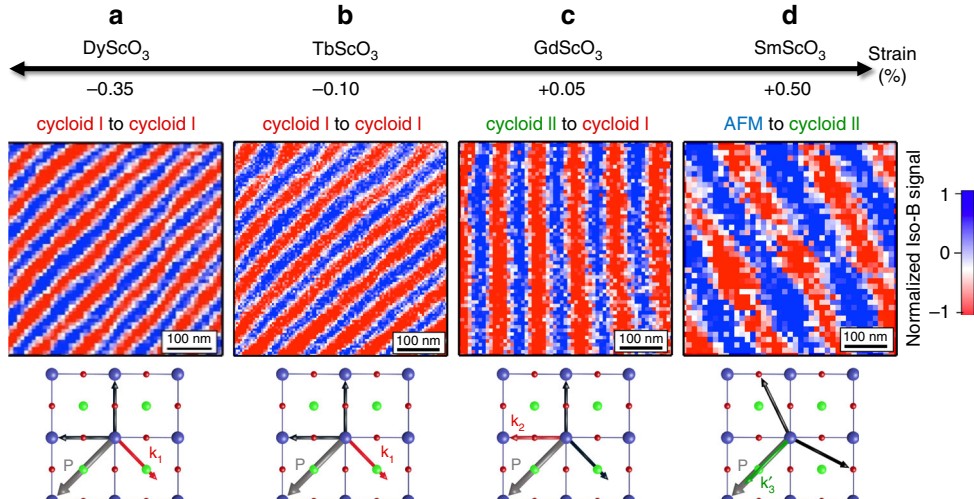

**Fig. 5 Magnetic textures in single ferroelectric domains as a function of strain. a–d** NV magnetometry images in single ferroelectric domains defined by PFM for BiFeO$_3$ thin films grown on DyScO$_3$ (**a**), TbScO$_3$ (**b**), GdScO$_3$ (**c**), and SmScO$_3$ (**d**). The corresponding strain values are depicted in the first row and the second row presents the evolution of the magnetic textures from striped domains to single ferroelectric domains. The propagation vector of the spin cycloid relative to the ferroelectric polarisation is sketched below each image. The symbol AFM in (**d**) stands for the pseudo-collinear antiferromagnetic order.

($\lambda_{surf} = \sqrt{2} \times \lambda$), giving rise to an intrinsic period of $\lambda = 65 \pm 2$ nm, close to the bulk value. These experiments on single domains suggest that strain primarily has an influence on the direction of the bulk-like cycloid propagation (in-plane for compressive and out-of-plane for tensile strains). In the case of BiFeO$_3$ films grown on SmScO$_3$ (+0.50%, Fig. 5d), the cycloid is observed to propagate in a direction almost parallel to the in-plane variant of polarisation. Considering the three vectors of each cycloid type (Fig. 3), this is only compatible with the $\mathbf{k}_3'$ propagation vector of cycloid II. In this case, we find an apparent cycloid period of $146 \pm 5$ nm leading to an intrinsic period of $84 \pm 3$ nm ($\lambda_{surf} = \sqrt{3} \times \lambda$). The enhanced period compared to the bulk value is here attributed to the significant tensile strain of BiFeO$_3$ films grown on SmScO$_3$ (Ref. [11].). In this latter example, we have demonstrated electric-field switching from a G-type antiferromagnetic order to a cycloidal state.

In this work, we have shown real-space evidence of multiple antiferromagnetic landscapes in BiFeO$_3$ epitaxial thin films. Depending on the strain level, bulk-like cycloids, exotic cycloids, and G-type collinear order are observed. The exotic cycloid is, rather unexpectedly, shown to exist for two very different strain states: one being compressive and the other tensile. Combining multiple scanning probe techniques, we provide direct correspondence between ferroelectric domains and complex antiferromagnetic textures. These local observations are supported by macroscopic resonant X-ray scattering on both types of cycloids. Although the cycloid is often not considered in the literature of BiFeO$_3$ thin films[27], our observations show that only the cycloidal state enables a full one-to-one correspondence between ferroic orders in the native striped-domains as well as in artificially-designed single domains. The electric field enables toggling either from one type of cycloid to another or from collinear to cycloidal states. More specifically, we are now able to electrically design single spin cycloids on demand with controlled propagation either in the plane or out of the film plane. This fully mastered magnetoelectric system is an ideal playground to investigate reconfigurable low-power antiferromagnetic spintronic[1,28,29] or magnonic[30] architectures at room temperature.

## Methods

**Sample fabrication**. BiFeO$_3$ thin films were grown by pulsed laser deposition on various substrates using a KrF excimer laser (248 nm) with a fluence of 1 J cm$^{-2}$. Prior to film growth, the scandate substrates (DyScO$_3$, TbScO$_3$, GdScO$_3$, SmScO$_3$) were ex-situ annealed for 3 h at 1000 °C under flowing oxygen. The SrTiO$_3$ substrate was chemically etched with a buffered HF solution before following the same annealing procedure. For all the samples, a SrRuO$_3$ bottom electrode (3–5 nm) was first grown at 660 °C under 0.2 mbar of oxygen pressure with a laser repetition rate of 5 Hz. The BiFeO$_3$ thin film (30–60 nm) was subsequently grown at the same temperature under 0.36 mbar of oxygen pressure and a repetition rate of 2 Hz. Following the growth of the bilayer, the samples were cooled down to room temperature under an oxygen pressure of 300 mbar.

**Structural characterisations**. The structural properties of the films were determined by X-ray diffraction (XRD) using a Panalytical Empyrean diffractometer equipped with a hybrid monochromator for Cu $K_{\alpha 1}$ radiation and a PIXcel3D detector. Full $2\theta$–$\omega$ XRD scans (not shown) indicate that all films are single phase with a monoclinic (001) orientation. To gain further insight into the elastic domains and strain of the films, we carried out reciprocal space maps (RSMs) around the (103), (013), (113), and ($\bar{1}\bar{1}$3) substrate peaks (Fig. 1f–j and Supplementary Fig. 1). The (110) orthorhombic scandates (XSO with X = Dy, Tb, Gd, Sm) are all described in a (001) monoclinic (which is only a slight correction from pseudo-cubic) notation for simplicity[31]. All the RSMs are consistent, with only two monoclinic ferroelastic variants of BiFeO$_3$ with the following epitaxial relationship: (001)BFO|| (001)XSO, [100]BFO|| [110]XSO (green) and (001)BFO|| (001)XSO, [100]BFO|| [1$\bar{1}$0]XSO (blue). The same epitaxial relationship is established for BiFeO$_3$ films grown on cubic (001)SrTiO$_3$ substrates. The BiFeO$_3$ thin films are fully strained by the substrates as indicated by the alignment of the in-plane reciprocal peaks with the (103) and (013) substrate peaks (Supplementary Fig. 1). The monoclinic cell parameters ($a_m, b_m, c_m, \beta$) of each BiFeO$_3$ film were calculated independently from the peak positions around the (113) and ($\bar{1}\bar{1}$3) RSMs of XSO. The strain values were then estimated by comparing the average in-plane lattice

parameter with the volume of the unit-cell as:

$$\varepsilon = \frac{\sqrt{\frac{a_m \times b_m}{2}} - \sqrt[3]{\frac{V}{2}}}{\sqrt[3]{\frac{V}{2}}}, \text{where } V = a_m \times b_m \times c_m \times \sin\beta$$

Considering the small deviation from the cubic unit cell, cell, throughout the manuscript, descriptions of the ferroelectric and magnetic properties are given in the pseudo-cubic perovskite lattice for simplicity.

**Piezoresponse force microscopy**. The experiments were conducted with an atomic force microscope (Nanoscope V multimode, Bruker) and two external lock-in detectors (SR830, Stanford Research) for the simultaneous acquisition of in-plane and out-of-plane responses. An external ac source (DS360, Stanford Research) was used to excite the SrRuO$_3$ bottom electrode at a frequency of 35 kHz while the conducting Pt-coated tip was grounded. We used stiff cantilevers (40 N m$^{-1}$) for accurate out-of-plane detection and softer ones (3-7 N m$^{-1}$) for the in-plane detection. In all the BiFeO$_3$ samples, the as-grown out-of-plane signal is homogeneous (Supplementary Fig. 2a) indicating a uniform out-of-plane component of polarisation pointing downwards, i.e. towards the SrRuO$_3$ bottom electrode. In Fig. 2a–e and Supplementary Figs. 2–5 and Supplementary Fig. 7, the phase shift between the in-plane and out-of-plane domains is 180 degrees and the phase scale is fixed at 360 degrees to avoid saturation of the image. Before designing artificial domains in the BiFeO$_3$ thin films, a radio frequency antenna and markers are defined by laser lithography and lift-off of a Au/Ti sputtered layer (Supplementary Fig. 5). These markers are typically less than 10 μm away from the antenna and are visible with an optical microscope. Optical microscopy allows for coarse repositioning, and maps provided by PFM measurements (including markers; Supplementary Fig. 5) are used to precisely relocate NV imaging.

**Scanning NV magnetometry**. Scanning-NV magnetometry was performed under ambient conditions with commercial all-diamond scanning-probe tips containing single NV defects (QNAMI, Quantilever MX). The tip was integrated into a tuning-fork based atomic force microscope (AFM) combined with a confocal microscope optimized for single NV defect spectroscopy. Magnetic fields emanating from the sample are detected by recording the Zeeman shift of the NV defect's electronic spin sublevels through optical detection of the electron spin resonance[17].

The scanning-NV magnetometer was operated in the dual-iso-**B** imaging mode by monitoring the signal $S = PL(v_2) - PL(v_1)$, corresponding to the difference of photoluminescence (PL) intensity for two fixed microwave frequencies, $v_1$ and $v_2$, applied consecutively at each point of the scan through a gold stripline antenna directly fabricated onto the BiFeO$_3$ sample (see the description before)[17]. Experiments were performed with a NV-to-sample distance of 60 nm and a bias magnetic field of 2 mT applied along the NV quantization axis. The standard error of the cycloid period measurement is limited by the calibration of the scanner.

**Resonant X-ray elastic scattering**. Resonant X-ray scattering measurements were performed at the Fe $L$ and O $K$ edges using the RESOXS diffractometer[32] at the SEXTANTS beamline[33] of the SOLEIL synchrotron. Data were collected using nearly fully circular left and right X-ray polarisations delivered by the HU44 Apple2 undulator located at the I14-M straight section of the storage ring.

**Microdiffraction**. The experiments were performed using a Bruker D8 Discover diffractometer with a high brilliance microfocus Cu rotating anode generator, hybrid Montel optics, a 20 μm diameter circular pinhole beam collimator, and an EIGER2 R 500 K area detector. No monochromator was used to maximize the flux from the microfocus lab source, leading to the characteristic $K_{\alpha 1,2}$ peak splitting. Prior to the microdiffraction experiments, a lithographically defined hard mask of 90 nm thick Au with 30 μm wide square openings was applied by sputtering and lift-off for precise alignment and orientation on the sample. Selected areas, written and pristine, with different domain wall densities were first analysed by PFM and subsequently by microdiffraction at the same area to obtain local structural information.

## Data availability
The data that support the findings of this study are available from the corresponding author upon request.

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

## Acknowledgements

We acknowledge support from the French Agence Nationale de la Recherche (ANR) through the PIAF and SANTA projects, the European Research Council (ERC-StG-2014, Imagine), the EU Quantum Flagship project ASTERIQS (820394) and the European Union's Horizon 2020 research and innovation programme under the Marie Sklodowska-Curie grant agreement No 846597. This work was supported by a public grant overseen by the ANR as part of the 'Investissement d'Avenir' programme (LABEX NanoSaclay, ref. ANR-10-LABX-0035). FG acknowledges the Graphene flagship 696656 and 785219. We also acknowledge the company QNAMI for providing all-diamond scanning tips containing single NV defects.

## Author contributions

V.G., S.F. and V.J conceived and coordinated the experiment. J.F. and C.C. prepared the samples. J.F. carried out the X-ray diffraction experiments and analysed the structural properties of the samples with D.S. and V.G. S.F. and F.G. patterned the microwave antennas and markers for repositioning. J.F., S.F. and V.G. performed the piezoresponse force microscopy experiments. A.H., W.A., A.F. and V.J. conducted the scanning NV magnetometry experiments. J.-Y.C., N.J. and M.V. performed the resonant X-ray scattering experiments. Y.A.B. conducted microdiffraction experiments. V.G. and S.F. wrote the paper with inputs from J.F., M.B., D.S. and V.J. All the authors discussed the data and commented on the paper.

## Competing interests

The authors declare no competing interests.
