## [Peer Review File · Nature Communications]

Reviewers' comments:

Reviewer #1 (Remarks to the Author):

This is an interesting paper, and the real space imaging of antiferromagnetic order of BFO, along with its strain engineering, is very valuable, and shed further insight into its magnetoelectric coupling. Nevertheless, I feel some clarifications are necessary to help the readers.

1. When introducing electric control of antiferromagnetic order at the beginning, earlier works on BFO deserve discussions, where local imaging has been demonstrated.
2. Throughout the manuscript, scale bar for PFM phase appear to be in the range from 0 to 360. If so, there appears to be no phase contrast then. Please double check the range.
3. Assigning PFM domain wall to be 71 instead of 109, is not completely supported by the data presented. For lateral PFM, I believe mapping under at least two orientation is necessary. Otherwise, I am not sure how authors can conclude that it is 71.
4. Can we spatially relate PFM and NV mapping? These two sets of mapping, as I understanding, do not spatially correlate with each other. Maybe a marker can be introduced?
5. Can authors explain the meaning of color contrast in NV mapping, blue and red?
6. How typical is Fig. 2k, where no correlation with polarization is seen? Is it common throughout that particular sample?
7. For Figure 5, I believe it is necessary to show NV mapping before and after poling. It is also important to show PFM before and after.
8. Supporting figures need better caption, i.e. more details.
- 9.

Reviewer #2 (Remarks to the Author):

The manuscript by A. Haykal et al. reported imaging of variety of antiferromagnetic (AF) spin textures in multiferroic BiFeO₃ (BFO) thin films tuned by strain and electrical field using combination of piezoresponse force microscopy and scanning probe NV magnetometry. The results show strong correlation between the spin configuration and polarization. The image reveals nicely the evolution of real space AF configuration such as collinear or different cycloid state as a function of strain and electrical field.

The author performed a very systematic study of magnetoelectric effect in BFO combining various different techniques and material parameters. The images presented in the paper are of high quality. The methods and conclusions in the paper are sound. AFs are considered promising candidate for next-generation GHz - THz electronic technologies. These material systems demonstrated in this paper paves the way for novel, low-power antiferromagnetic spintronic devices at room temperature. I therefore recommend this paper for publication.

Reviewer #1:

This is an interesting paper, and the real space imaging of antiferromagnetic order of BFO, along with its strain engineering, is very valuable, and shed further insight into its magnetoelectric coupling. Nevertheless, I feel some clarifications are necessary to help the readers.

We thank the Reviewer for mentioning that our paper is “interesting” and “shed[s] further insight into” the magnetoelectric coupling in BiFeO₃. Below, we address point by point the specific issues raised by the Reviewer.

1. When introducing electric control of antiferromagnetic order at the beginning, earlier works on BFO deserve discussions, where local imaging has been demonstrated.

The electric control of antiferromagnetism and microscopic correlations between the ferroelectric and the antiferromagnetic domain structures were reported in the seminal work of Zhao *et al.* (Nature Mater. 5, 823 (2006)). Following this article and thanks to the interfacial magnetic exchange, correlations between the magnetic textures of a Co_{0.9}Fe_{0.1} layer grown on BiFeO₃ and the ferroelectric domains of the multiferroic (Trassin *et al.*, Phys. Rev. B 87, 134426 (2013)), led to the electric control of the magnetization of Co_{0.9}Fe_{0.1} (Heron *et al.*, Nature 516, 370–373 (2014)). We assume this is the earlier work the Reviewer is referring to. We point out that in all these previous works, the magnetic structure of BiFeO₃ is considered as a (canted G-type) pseudo-collinear antiferromagnetic order and the electric control of magnetism is systematically interpreted within this framework. We believe that these interpretations are somehow incomplete as the spin cycloid should be present in the BiFeO₃ thin films considered earlier, as reported for very similar heterostructures in our previous work (Gross *et al.*, Nature 549, 252 (2017)) as well as in this manuscript. The canted magnetic moment is then not constant in each domain but follow the cycloid as a spin density wave. Nevertheless, following the Reviewer’s suggestion, we added a sentence in the introduction mentioning these earlier works in the manuscript: “The seminal work of Zhao *et al.* showed promise for the electric control of the antiferromagnetic order in BiFeO₃ thin films”.

2. Throughout the manuscript, scale bar for PFM phase appear to be in the range from 0 to 360. If so, there appears to be no phase contrast then. Please double check the range.

The Reviewer is right when mentioning that a PFM phase contrast of 360 degrees would mean no contrast. We actually get a phase shift of 180 degrees for the in-plane and out-of-plane PFM phase images, as we would expect in the case of in-plane and out-of-plane signals that are free of parasitic contributions. To avoid any saturation of the images, we and others usually present the PFM phase images with a vertical scale of 360 degrees. We added this information to the Methods.

3. Assigning PFM domain wall to be 71 instead of 109, is not completely supported by the data presented. For lateral PFM, I believe mapping under at least two orientation is necessary. Otherwise, I am not sure how authors can conclude that it is 71.

The Reviewer is absolutely right and we checked that the domain walls were 71-degree domain walls by doing vectorial PFM (cantilever aligned in different in-plane directions). This is illustrated in **Figure R1** where the out-of-plane phase signal indicates that all domains have downward polarisation. Using three different in-plane configurations, we can reveal that only two in-plane polarisation variants exist with a corresponding angle of 71 degrees (see the figure caption for details). We added this Figure as a Supplementary Figure. Alternatively, we usually do the in-plane PFM measurements in two positions: position 1 and position 4 where the cantilever is parallel to the stripes. Two in-plane PFM phase contrasts (bright and dark) in position 1 and a single one in position 4 correspond to solely 71-degree domain walls with two polarisation variants. For the sake of simplicity, we only display PFM images

acquired in the first configuration of **Figure R1** where the contrasts (bright and dark) correspond to two alternated polarisation variants, separated by 71-degree domain walls. This observation is valid for all the striped domain structures displayed in Figure 2 of the manuscript.

Figure R1: Vectorial PFM on BiFeO₃ thin films grown on SrRuO₃/DyScO₃(110). In position 1, the cantilever is perpendicular to the ferroelectric stripes. The out-of-plane PFM phase is homogeneous and bright, indicating only downward polarisation variants. The in-plane PFM phase shows alternating bright and dark domains with equal amplitudes. As sketched on the right panel, this could correspond to (P₁, P₄) variants in the bright regions and (P₂, P₃) variants in the dark regions. In position 2, the cantilever is parallel to (P₂, P₄), thus these two variants do not respond. The in-plane PFM phase and amplitude show that only one family of domains responds and its phase signal is thus bright. This signal corresponds to the P₃ variant (right of the cantilever). In position 3, the cantilever is parallel to (P₁, P₃). The in-plane PFM phase and amplitude show that only one family of domains responds and its phase signal is therefore dark. This signal corresponds to the P₄ variant (left of the cantilever) and the P₃ variant does not respond as it is parallel to the cantilever. Putting all this information together allows us to conclude that the striped-domain structure then corresponds to alternated P₃ and P₄ domains with 71-degree domain walls. All the PFM images are 2.5×2.5 μm². The dashed red line emphasizes the complementarity between each signal in the three different positions.

4. Can we spatially relate PFM and NV mapping? These two sets of mapping, as I understanding, do not spatially correlate with each other. Maybe a marker can be introduced?

Before designing artificial domains in the BiFeO₃ thin films, a radio frequency antenna as well as markers are defined by laser lithography and lift-off of a Au/Ti sputtered layer (**Figure R2**). These markers are typically less than 10 micrometres away from the antenna and are visible with an optical microscope. Optical microscopy allows for coarse repositioning, and maps provided by PFM measurements (including markers, **Figure R2**) are used to precisely relocate NV imaging. We added this information to the Methods and as a Supplementary Figure.

Figure R2: Antenna and markers defined by laser lithography on the BiFeO₃ samples. (top) Optical microscope image. (bottom) 17 × 8.5 μm² PFM images in the scanned area defined by the dashed yellow square.

5. Can authors explain the meaning of color contrast in NV mapping, blue and red?

The NV images are acquired in the iso-B mode, as detailed in the Methods section. This means that each pixel is measured for two different rf frequencies around the electron spin resonance of the NV center. It should be noted that a small (bias) magnetic field is preliminarily applied in order to lift the degeneracy between the +1 and -1 spin states. The iso-B image is then the difference of photoluminescence collected for these two frequencies. The blue and red indicate a positive and negative shift of the resonance, corresponding to a positive or negative magnetic stray field projected along the NV center axis.

6. How typical is Fig. 2k, where no correlation with polarisation is seen? Is it common throughout that particular sample?

All the PFM and NV magnetometry images displayed in Figure 2 (including Figure 2k) are representative images that were observed for various locations in the BiFeO₃ films. The lack of correlation between the ferroelectric domains and the magnetic textures for BiFeO₃ films grown on SmScO₃(110) substrates is confirmed for different locations of the film, as visible in **Figure R3**. Indeed, it appears that when the spin cycloid is destroyed, there are multiple variants of antiferromagnetic domains in each ferroelectric domain. We added this figure as a Supplementary Figure.

Figure R3: NV magnetometry images at different locations of the BiFeO₃ film grown on SrRuO₃/SmScO₃(110)

7. For Figure 5, I believe it is necessary to show NV mapping before and after poling. It is also important to show PFM before and after.

Showing NV imaging before and after defining single ferroelectric domains would be equivalent to merging Figure 2 and Figure 5. We have tried this option and believe this would make the figure too heavy. As an alternative, we can provide more details where we show the PFM images corresponding to the NV images of Figure 5 (see **Figure R4**). We added this figure as a Supplementary Figure.

Figure R4: (top) In-plane PFM phase images of written areas and (bottom) corresponding NV images for BiFeO₃ films grown on (a) DyScO₃, (b) TbScO₃, (c) GdScO₃, and (d) SmScO₃ substrates. The dashed squares in the PFM images show the sizes of the corresponding NV images.

8. Supporting figures need better caption, i.e. more details.

Following the Reviewer's comments, we have provided additional details to the Supplementary Figure captions.

Reviewer #2 (Remarks to the Author):

The manuscript by A. Haykal et al. reported imaging of variety of antiferromagnetic (AF) spin textures in multiferroic BiFeO₃ (BFO) thin films tuned by strain and electrical field using combination of piezoresponse force microscopy and scanning probe NV magnetometry. The results show strong correlation between the spin configuration and polarisation. The image reveals nicely the evolution of real space AF configuration such as collinear or different cycloid state as a function of strain and electrical field.

The author performed a very systematic study of magnetoelectric effect in BFO combining various different techniques and material parameters. The images presented in the paper are of high quality. The methods and conclusions in the paper are sound. AFs are considered promising candidate for next-generation GHz - THz electronic technologies. These material systems demonstrated in this paper paves the way for novel, low-power antiferromagnetic spintronic devices at room temperature. I therefore recommend this paper for publication.

REVIEWERS' COMMENTS:

Reviewer #1 (Remarks to the Author):

The authors adequately addressed the issues raised, and I recommend its publication.